# Admission Red Blood Cell Distribution Width and Mean Platelet Volume as Predictors of Mortality in the Pediatric Intensive Care Unit: A Five-Year Single-Center Retrospective Study

**DOI:** 10.3390/jcm14113839

**Published:** 2025-05-29

**Authors:** Kanokkarn Sunkonkit, Chatree Chai-adisaksopha, Rungrote Natesirinilkul, Phichayut Phinyo, Konlawij Trongtrakul

**Affiliations:** 1Division of Pulmonary and Sleep Medicine, Department of Pediatrics, Faculty of Medicine, Chiang Mai University, Chiang Mai 50200, Thailand; kanokkarn.sun@cmu.ac.th; 2Center for Clinical Epidemiology and Clinical Statistics, Faculty of Medicine, Chiang Mai University, Chiang Mai 50200, Thailand; chatree.chai@cmu.ac.th (C.C.-a.); phichayut.phinyo@cmu.ac.th (P.P.); 3Department of Biomedical Informatics and Clinical Epidemiology (BioCE), Faculty of Medicine, Chiang Mai University, Chiang Mai 50200, Thailand; 4Division of Hematology, Department of Internal Medicine, Faculty of Medicine, Chiang Mai 50200, Thailand; 5Division of Hematology and Oncology, Department of Pediatrics, Faculty of Medicine, Chiang Mai University, Chiang Mai 50200, Thailand; rungrote.n@cmu.ac.th; 6Division of Pulmonary, Critical Care, and Allergy, Department of Internal Medicine, Faculty of Medicine, Chiang Mai University, Chiang Mai 50200, Thailand

**Keywords:** red blood cell distribution width, mean platelet volume, intensive care unit, critically ill children, PICU mortality, children

## Abstract

**Background/Objectives:** Red blood cell distribution width (RDW) and mean platelet volume (MPV) are well-established prognostic biomarkers across various medical conditions. However, their role in predicting mortality among critically ill pediatric patients remains unclear. This study investigates the association between RDW, MPV, and 28-day mortality in pediatric intensive care unit (PICU) patients. **Methods**: This retrospective cohort study analyzed data from children aged 1 month to 18 years who were admitted to the PICUs at Chiang Mai University Hospital for ≥24 h between January 2018 and December 2022. The primary outcome was 28-day PICU mortality. A log-binomial regression analysis was conducted to assess the association of RDW and MPV with 28-day PICU mortality, adjusting for age, sex, mechanical ventilation, vasoactive drug use, continuous renal replacement therapy, and multiorgan failure. **Results**: A total of 580 PICU patients were included, 55.3% male, with a median age of 5.9 (IQR: 4.7–10.4) months. The 28-day PICU mortality rate was 9.8% (57/580). Respiratory failure and acute respiratory distress syndrome were the most common admission diagnoses (72.1%). Elevated RDW (≥15%) and MPV (≥10 fL) were independently associated with increased 28-day PICU mortality (adjusted RR: 2.73, 95% CI: 1.45–5.13 and 2.38, and 95% CI: 1.43–3.93, respectively). Both markers demonstrated high negative predictive values (RDW: 96.0% and MPV: 94.6%). **Conclusions**: Elevated RDW (≥15%) and MPV (≥10 fL) were independently associated with increased 28-day PICU mortality. These findings highlight their potential utility as accessible and cost-effective biomarkers for early risk stratification in critically ill pediatric patients.

## 1. Introduction

The complete blood count (CBC) is an affordable and widely accessible test that evaluates red blood cell (RBC), white blood cell (WBC), and platelet characteristics. Numerous studies have shown associations between RBC and platelet parameters—such as red cell distribution width (RDW) and mean platelet volume (MPV)—and inflammatory stages, suggesting their potential as predictors of mortality in the intensive care unit (ICU) [1,2,3,4,5,6,7,8]. These parameters are easy to implement in routine clinical practice and can help identify high-risk critically ill pediatric patients who may benefit from prioritized ICU or intermediate care admission, particularly in resource-limited environments.

RDW has traditionally been utilized as an ancillary marker to aid in differentiating the causes of anemia [9]. However, emerging evidence has identified RDW as a significant prognostic indicator across various systemic diseases, with increasing recognition of its association with mortality in critically ill patients [9,10,11,12,13,14]. MPV, derived by dividing the platelet-crit (PCT) by the platelet count [15,16], reflects the average size of circulating platelets and is readily measured through automated hematology analyzers. Elevated MPV is frequently observed in thrombotic and inflammatory states, reflecting heightened platelet activation and function [17,18].

Both RDW and MPV are readily accessible laboratory parameters that encapsulate underlying pathophysiological changes common in critical illness. RDW, an indicator of anisocytosis, has been linked to systemic inflammation, oxidative stress, and impaired erythropoiesis—conditions prevalent among critically ill children and associated with adverse clinical outcomes [19,20]. Similarly, MPV serves as a surrogate marker of platelet activation, which plays a central role in the inflammatory cascade and the development of microthrombotic complications seen in severe infections and systemic inflammatory states [18]. Prior studies have demonstrated that elevated MPV levels correlate with poor prognosis in both adult and pediatric critically ill populations [8]. Collectively, these mechanisms provide a compelling biological rationale for evaluating RDW and MPV as potential prognostic indicators of mortality risk in the pediatric intensive care unit (PICU).

Due to the accessibility, affordability, and widespread availability of CBC testing across all levels of healthcare, CBC parameters hold promise as practical indicators for predicting mortality risk. If these routinely measured biomarkers can reliably forecast mortality, they could form the foundation for a new scoring system, providing a practical alternative to more costly and complex novel biomarkers that are less accessible in routine clinical practice [21,22]. However, most research has focused on adult patients, with relatively few studies examining RDW and MPV in children, especially within critically ill pediatric populations. Therefore, this study aims to determine and explain the association between RDW and MPV and 28-day ICU mortality in critically ill children.

## 2. Materials and Methods

### 2.1. Study Design and Setting

This retrospective study was conducted on critically ill pediatric patients admitted to the two PICUs at the Faculty of Medicine, Chiang Mai University, Thailand, from January 2018 to December 2022. The study protocol received approval from the Research Ethics Committee of the Faculty of Medicine (approval no. 166/2566, dated 10 May 2023). Conducted in line with the Declaration of Helsinki’s ethical principles for medical research involving human subjects, the study was exempt from informed consent due to its minimal risk design, with all data anonymized for analysis.

### 2.2. Eligibility Criteria

#### 2.2.1. Inclusion Criteria

Critically ill children aged 1 month to 18 years admitted to our PICUs were included in the study. Eligible patients had a CBC upon admission and remained in the ICU for at least 24 h. Each subsequent readmission was considered a new episode.

#### 2.2.2. Exclusion Criteria

Patients were excluded if they had received an RBC transfusion within 21 days or a platelet transfusion within 7 days before the PICU admission. Additionally, patients with conditions that could affect RDW and platelet indices were excluded, including those with beta-thalassemia major, beta-thalassemia/hemoglobin E, hemoglobin H disease, congenital hemolytic anemia (e.g., RBC membrane defects), congenital macrothrombocytopenia, or myeloproliferative disorders.

### 2.3. Data Collection

We collected data on each participant’s demographics, including age, gender, and body mass index (BMI), along with information about their ICU admission, comorbidities, and interventions. Specifically, we documented the primary reason for ICU admission, the use of mechanical ventilation (MV), the administration of vasoactive medications, continuous renal replacement therapy (CRRT), the presence of multi-organ dysfunction (MOD), the severity of illness measured by the Pediatric Index of Mortality (PIM-2) scores, and the duration of MV support, vasoactive drug administration, and ICU stay. Laboratory data within the first 24 h of ICU admission were collected, including CBC, blood urea nitrogen (BUN), creatinine, erythrocyte sedimentation rate (ESR), C-reactive protein (CRP), lactate, and liver function tests. CBC analyses were performed using the Sysmex XN-9000^®^ analyzer (Sysmex Co., Kobe, Japan).

### 2.4. Outcome

Twenty-eight-day PICU mortality was defined as death occurring within 28 days of admission to the PICU. This 28-day period represents a complete 4-week interval, which aligns with standard clinical follow-up practices in the PICU setting, ensures uniformity in data collection intervals, and corresponds to a widely accepted timeframe for outcome assessment in critical care research.

### 2.5. Study Size Estimation

The sample size for this study was calculated using independent two-sample comparisons of the means. Data on RDW (17.7 ± 5.0% in 90 survivors vs. 21.0 ± 6.1% in 11 non-survivors) and MPV (8.5 ± 1.1% in 310 survivors vs. 9.5 ± 1.6% in 15 non-survivors) were derived from the studies by Sachdev et al. [1] and Kim et al. [15], respectively. Based on an alpha level of 0.05 (two-sided) and a power of 0.80, the estimated sample sizes for RDW and MPV were 435 and 330 cases, respectively. To account for 25% potential data loss during collection, a total sample size of 580 was required in our study.

### 2.6. Statistical Analysis

Categorical variables were presented as frequencies and percentages. Continuous variables were presented as mean and standard deviation (SD) or median and interquartile range (IQR), as appropriate. Causal inference of RDW and MPV on 28-day PICU mortality was assessed using directed acyclic graphs (DAGs) [23]. We identified the minimally sufficient adjustment set for RDW by controlling for age, sex, the use of MV, vasoactive agents, CRRT, and MOD. For MPV, the adjustment set included sex, the use of MV, vasoactive agents, CRRT, and MOD. Log-binomial regression analyses were conducted to evaluate the association between RDW and MPV with 28-day PICU mortality, and results were reported as risk ratios (RR). RDW and MPV were modeled as linear and quadratic terms to evaluate both linear and non-linear associations with 28-day mortality. For practical clinical application, both variables were subsequently dichotomized based on optimal sensitivity and specificity thresholds: RDW (<15% vs. ≥15%) [2,24,25,26] and MPV (<10 fL vs. ≥10 fL) [8,27,28]. The discriminative ability of RDW and MPV in predicting 28-day PICU mortality was assessed using the area under the receiver operating characteristic (AuROC) curve, with corresponding sensitivity, specificity, positive predictive value (PPV), and negative predictive value (NPV) reported. All statistical analyses were performed using STATA version 17 (StataCorp, College Station, TX, USA). A two-tailed *p*-value of <0.05 was considered statistically significant.

## 3. Results

A total of 900 children admitted to the PICU were initially considered for this study. Of these, 320 children were excluded due to recent transfusions: 208 children had received RBC transfusions within the past 21 days, 17 children had undergone platelet transfusions in the previous 7 days, and 95 children had received both, as shown in Figure 1. Consequently, 580 children were included in the final analysis. The 28-day PICU mortality rate was 9.8% (57/580).

Table 1 presents the baseline characteristics, comparing survivors and non-survivors. Non-survivors were generally younger and had a higher proportion of males than survivors, with a median age of 5.8 (IQR: 4.7–8.8) months compared to 6.0 (IQR: 4.8–10.4) months in survivors (*p* = 0.689). Males comprised 66.7% of non-survivors versus 55.4% of survivors (*p* = 0.070).

The primary reasons for ICU admission were respiratory failure and acute respiratory distress syndrome (ARDS) (72.1%), congestive heart failure (22.4%), post-operative care (19.5%), and sepsis/septic shock (17.1%), respectively. Intubation was required in 74.5% (432 children) and vasoactive medications were administered to 37.4% (217 children) within the first 24 h of admission. Interestingly, 44.6% (259 children) had MOD, and 4.3% (25 children) commenced CRRT. The non-survivor group demonstrated a significantly prolonged duration of MV (*p* < 0.001) and extended use of vasoactive medications (*p* < 0.001) compared to the survivor group. Additionally, the non-survivors exhibited significantly higher levels of RDW (*p* < 0.001), mean corpuscular volume (*p* = 0.014), MPV (*p* < 0.001), and platelet distribution width (PDW) (*p* < 0.001), along with a notably lower platelet count (*p* < 0.001). A comprehensive summary of PICU outcomes and laboratory parameters is presented in Table 2.

RDW was significantly associated with 28-day PICU mortality in both linear and quadratic regression models. However, a comparison of model fit revealed no statistically significant difference (*p* = 0.46), supporting the selection of the more parsimonious linear model. In this model, RDW was identified as a significant predictor of 28-day PICU mortality, with a crude RR of 1.25 (95% CI: 1.19–1.33, *p* < 0.001). After adjustment for age, sex, the use of MV, vasoactive agents, CRRT, and MOD, the association remained significant (adjusted RR: 1.19; 95% CI: 1.13–1.25; *p* < 0.001), with an AuROC of 0.73 (95% CI: 0.66–0.81). The causal diagram, the association between elevated RDW levels and 28-day PICU mortality, and the predictive performance of RDW are illustrated in Figure 2A and Figure 3A,B and Table 3, respectively.

To determine the optimal cut-off value for RDW, a receiver operating characteristic (ROC) curve analysis was performed. RDW was dichotomized at <15% and ≥15%. An RDW ≥15% was identified as a significant predictor of 28-day PICU mortality, with a crude RR of 3.82 (95% CI: 2.02–7.23, *p* < 0.001) and adjusted RR (aRR) of 2.73 (95% CI: 1.45–5.13, *p* = 0.002) after accounting for relevant covariates including age, sex, use of MV, vasoactive agent use, CRRT, and MOD (Table 3). The RDW threshold of 15% demonstrated optimal performance in predicting 28-day PICU mortality, with corresponding sensitivity, specificity, PPV, and NPV of 80.7%, 50.9%, 15.2%, and 96.0%, respectively. The discriminative ability of RDW ≥15% for predicting 28-day PICU mortality was reflected by an AuROC of 0.66 (95% CI: 0.60–0.71) (Table 4).

The association between MPV and 28-day PICU mortality was also evaluated using both linear and quadratic regression models. MPV demonstrated a significant association with 28-day PICU mortality in both models. However, a comparison of model fit showed no statistically significant difference between the two approaches (*p* = 0.76), supporting the use of the more parsimonious linear model as well. In the linear regression analysis, MPV was identified as a significant predictor of 28-day PICU mortality, with a crude RR of 1.54 (95% CI: 1.25–1.89; *p* < 0.001). This association remained significant after adjusting for age, sex, the use of MV, vasoactive agent use, CRRT, and MOD, with an aRR of 1.34 (95% CI: 1.10–1.64; *p* = 0.003). The AuROC for the linear model was 0.65 (95% CI: 0.56–0.73). The causal diagram, the association between elevated MPV and 28-day mortality, and the predictive performance of MPV are presented in Figure 2B and Figure 4A,B and Table 3.

ROC curve analysis identified an optimal MPV threshold of ≥10 fL. MPV ≥ 10 fL was significantly associated with increased 28-day PICU mortality, with a crude RR of 3.04 (95% CI: 1.79–5.15; *p* < 0.001) in univariable analysis and an aRR of 2.38 (95% CI: 1.43–3.93; *p* = 0.001) after multivariable adjustment (Table 3). This cut-off demonstrated a sensitivity of 66.7%, specificity of 63.3%, PPV of 16.5%, NPV of 94.6%, and an AuROC of 0.65 (95% CI: 0.58–0.71). The predictive performance metrics of MPV ≥ 10 fL threshold are summarized in Table 4.

The PIM-2 score was associated with 28-day PICU mortality, yielding an AuROC of 0.59 (95% CI: 0.51–0.68). Interestingly, RDW demonstrated superior predictive performance for 28-day PICU mortality compared to the PIM-2 score (*p* = 0.015), whereas MPV showed no statistically significant difference when compared with PIM-2 score (*p* = 0.38).

## 4. Discussion

Our study underscores the significant association between elevated RDW and MPV levels, measured within 24 h of PICU admission, and 28-day PICU mortality. Although the AuROC analyses for RDW and MPV, as standalone predictors, indicated only moderate clinical utility when applying a single cut-off point, their discriminative performance improved markedly when patients were stratified into low- and high-risk groups. Specifically, RDW and MPV thresholds of 15% and 10 fL, respectively, yielded high NPV for 28-day PICU mortality, at 96.0% for RDW and 94.6% for MPV.

Importantly, as both RDW and MPV are routinely obtained from a standard CBC, they represent cost-effective and widely accessible tools for early risk stratification. These parameters might be integrated with other clinical variables to develop more comprehensive predictive models for mortality in critically ill pediatric patients, thereby supporting more informed clinical decision-making and potentially improving patient outcomes.

Our results highlighted that RDW measured within 24 h of PICU admission may serve as an early indicator for predicting mortality in critically ill pediatric patients. Recognizing these high-risk patients promptly could enable timely interventions and management strategies, potentially improving clinical outcomes and optimizing resource utilization. Importantly, the most notable advantages of RDW as a practical clinical marker are its cost-effectiveness and widespread availability, as it is readily obtained from a routine CBC test. This finding aligns with previous studies, including pediatric [1,2,5,13,14,26] and adult patients [4,10,11,12,29,30,31].

In the context of critically ill pediatric patients, previous studies have demonstrated a significant association between elevated RDW and adverse outcomes in the PICU setting. Sachdev et al. found that an RDW level of ≥18.6% at admission, along with persistently elevated values during hospitalization, was linked to increased mortality and prolonged PICU stay [1]. Similarly, Kim et al. categorized RDW into three groups (Group 1: <14.5%; Group 2: 14.5–16.5%; and Group 3: >16.5%) and reported that RDW served as an independent risk factor for PICU mortality.

It was also highlighted that incorporating RDW into the Pediatric Logistic Organ Dysfunction-2 (PELOD-2) scoring system significantly enhanced its predictive performance for mortality in critically ill children [2]. Additionally, Ramby et al. utilized multivariable logistic regression analysis to examine the relationship between RDW and clinical outcomes in 596 pediatric patients admitted to the PICU. It revealed that RDW measured on the first day of PICU admission was independently associated with mortality (OR 1.25, 95% CI 1.09–1.43), and the AUROC curves for RDW in predicting PICU length of stay >48 h and mortality were 0.61 (95% CI 0.56–0.66) and 0.65 (95% CI 0.55–0.75), respectively. A cut-off RDW value of >15.7% identified patients with a 78% risk of prolonged LOS and a 12.9% risk of mortality (*p* < 0.001) [26]. Consistently, our study demonstrates that elevated RDW is a reliable prognostic marker, offering comparable predictive value for PICU mortality.

The biological mechanisms linking RDW to mortality remain incompletely understood. However, inflammation and oxidative stress are thought to play key roles by disrupting erythrocyte homeostasis [30,32]. Inflammatory cytokines disrupt erythropoiesis by impairing erythrocyte maturation and accelerating cellular destruction or senescence. This process results in a premature release of immature reticulocytes into the peripheral circulation and delays the clearance of dysfunctional erythrocytes, ultimately leading to an increased RDW [19].

Elevated RDW levels in sepsis may also be associated with systemic inflammation. Proinflammatory cytokines, such as tumor necrosis factor-α (TNF-α), interleukin (IL)-6, and IL-1β, are believed to impair RBC maturation and shorten their lifespan [33,34]. Moreover, reactive oxygen species can reduce erythrocyte lifespan and promote the premature release of immature reticulocytes into the circulation [35]. In critically ill patients with cardiovascular disease, heightened inflammatory responses and oxidative stress disrupt iron metabolism, impair bone marrow hematopoiesis, and hinder erythrocyte maturation, collectively contributing to elevated RDW [30].

Furthermore, the excessive production of catecholamines and angiotensin leads to a cascade of adverse effects, including vasoconstriction, thrombosis, microcirculatory dysfunction, and tissue perfusion deficits. These processes result in ischemia and hypoxia, which further impair bone marrow function and disrupt erythropoietin activity—a key hormone regulating erythrocyte production, maturation, and survival [30]. These intricate pathophysiological interactions position RDW as a potential marker reflecting disease severity and poor prognosis in critically ill patients.

Our study demonstrates that an MPV ≥ 10 fL is significantly associated with 28-day PICU mortality (*p* = 0.001). This finding aligns with previous research indicating MPV as a prognostic marker across diverse populations, including adults, neonates, and pediatric patients [3,15,36,37,38]. MPV has been recognized as a marker of inflammation, disease severity, and the efficacy of anti-inflammatory treatment in chronic inflammatory disorders [15,17,18,36].

Elevated MPV has been linked to adverse clinical outcomes in several studies. Cai et al. conducted a retrospective cohort study involving 73 preterm neonates with sepsis to evaluate the prognostic value of MPV and RDW [3]. Their findings revealed that MPV and RDW were independent predictors of poor prognosis in preterm neonates with sepsis, with an OR = 3.226, *p* = 0.017, and OR = 2.058, *p* = 0.019, respectively. ROC analysis demonstrated an AUROC of 0.738 for MPV, 0.768 for RDW, and 0.854 for the combined MPV-RDW model, suggesting superior predictive accuracy when both indices were considered together.

Importantly, MPV levels were significantly higher in non-survivors compared to survivors, further supporting its prognostic value. In a prospective study by Kim et al., which included 345 adult patients admitted to the emergency department with severe sepsis and/or septic shock, changes in MPV within the first 72 h emerged as an independent predictor of 28-day mortality (hazard ratio (HR) = 1.44; 95% CI = 1.01–2.06; *p* = 0.044) after adjusting for confounding factors [15]. Their findings highlight the importance of continuous MPV monitoring during the early critical phase to stratify mortality risk effectively. Similarly, Zhang et al. examined 261 adult ICU patients (204 survivors and 57 non-survivors) and identified MPV >11.3 fL as an independent risk factor for mortality (*p* = 0.023) after adjusting for clinical variables [39].

Conversely, Ye et al. conducted a retrospective study involving 2319 mechanically ventilated children and found that platelet volume indices, including MPV, were not independently associated with mortality at admission [40]. However, temporal trends in MPV and PDW differed between survivors and non-survivors over a one-week observation period, suggesting that dynamic changes rather than baseline values may hold greater predictive value.

A systematic review and meta-analysis by Tajarernmuang et al. further explored the association between MPV and mortality in critically ill patients [36]. While initial MPV values were not consistently predictive of mortality, subsequent measurements beyond the third day showed potential prognostic utility. This review also highlighted significant heterogeneity across studies, which may limit the generalizability of findings.

Collectively, these studies underscore the evolving understanding of MPV as a prognostic marker. While baseline MPV may have limitations in certain populations, serial measurements and dynamic changes over time appear to offer more reliable insights into mortality risk. Our findings contribute to this growing body of evidence, emphasizing the potential role of MPV in risk stratification and clinical decision-making in critically ill pediatric patients.

The pathophysiologic mechanisms underlying the association between elevated MPV and mortality remain incompletely understood. However, increased MPV may signal a hypercoagulable state, heightened inflammatory response, and oxidative stress [3,40,41]. Previous studies have suggested that MPV serves as an inflammatory marker, with elevated levels correlating with active inflammatory diseases [42]. Emerging evidence indicates that MPV reflects both proinflammatory and prothrombotic conditions, mediated by thrombopoietin and various inflammatory cytokines, including IL-1, IL-3, IL-6, and TNF-α [18]. These inflammatory and thrombotic processes may influence platelet production, activation, and consumption, all of which are captured in routine blood analyses through MPV assessment.

Additionally, MPV is affected by platelet aging and the dynamic balance between platelet production and destruction. The degree of inflammation and fluctuations in MPV seem to be closely interrelated across various clinical conditions, although the exact impact remains controversial. Notably, because of the strong inverse correlation between platelet count and MPV observed in healthy individuals, trends in platelet count should always be considered when interpreting MPV values [18]. Moreover, Huczek et al. demonstrated that patients with elevated MPV at admission are at higher risk for thrombotic events, further supporting its prognostic value in critically ill populations [43]. Among the proposed mechanisms, a heightened inflammatory response in critically ill children appears to be the most plausible explanation for the observed association between increased MPV and mortality.

Moreover, elevated RDW and MPV values were significantly associated with mortality in the PICU in our study. However, it is essential to interpret these findings within the clinical context. Patients who required prolonged MV or vasoactive support demonstrated higher RDW and MPV levels, which likely reflect the underlying severity of their critical illness rather than indicating that these hematologic parameters are independent prognostic factors. Conditions such as ARDS and septic shock are commonly accompanied by systemic inflammatory responses, which can induce bone marrow stress, ineffective erythropoiesis, and thrombopoiesis abnormalities, thereby elevating RDW and MPV values [44,45]. Thus, these laboratory abnormalities may also serve as markers of severe inflammatory burden. Our findings underscore the importance of considering RDW and MPV within the broader clinical picture, integrating these parameters alongside established indicators of illness severity to guide risk stratification and clinical decision-making in PICU.

Notably, RDW demonstrated better discriminatory ability for predicting 28-day PICU mortality in our cohort compared to the PIM-2 score. In contrast, MPV did not exhibit a statistically significant difference in discriminative performance when compared with the PIM-2 score. Nevertheless, MPV presents a more practical advantage, as it is easier to obtain and interpret, relying on a single parameter rather than the multiple variables required for calculating the PIM-2 score. However, further validation studies are warranted to confirm these findings.

Our study has some limitations. First, this research was conducted at a single tertiary care center, potentially limiting the generalizability of our findings. Further multicenter studies may be warranted to elucidate this hypothesis. Second, residual confounding factors may have impacted the validity of our findings. Specifically, conditions such as thalassemia and iron deficiency anemia, which are highly prevalent in Thai children, particularly in Northern Thailand, may have influenced the results. The high prevalence of these conditions limited our ability to adequately adjust for them as potential confounders. As a result, we were unable to determine whether the observed associations with RDW were independent of these underlying hematologic conditions. Nonetheless, existing literature consistently demonstrates that chronic anemia and iron deficiency anemia in pediatric populations are associated with increased mortality and poor prognosis, further supporting the relevance of these factors in the context of pediatric critical illness [46,47,48]. Third, RDW and MPV elevations may reflect the severity of underlying disease states, such as ARDS and septic shock, which are themselves associated with high mortality rates and require interventions like MV and vasoactive support. These factors may confound the observed associations, as both RDW and MPV could act as surrogate markers rather than independent predictors of mortality.

Finally, the lack of long-term follow-up data presents another limitation, as our dataset did not include information on post-discharge outcomes. Despite these constraints, our study underscores the potential clinical utility of RDW and MPV—two hematological markers that are inexpensive, widely available, and routinely included in CBC testing across both primary and advanced healthcare settings. These parameters could serve as valuable tools for the early identification of high-risk patients, guiding clinical decision-making, and optimizing resource allocation in PICU care. To address these limitations and strengthen the clinical applicability of our findings, future multicenter prospective studies are warranted. Such research could enhance the precision and integration of RDW and MPV into standardized mortality risk assessment tools in PICU settings.

## 5. Conclusions

Our study demonstrates an independent association of elevated RDW and MPV with 28-day PICU mortality, highlighting their potential influence on patient outcomes. While their individual predictive value was moderate, their utility improved in risk stratification, with RDW ≥ 15% and MPV ≥ 10 fL showing high negative predictive value. Due to their affordability, accessibility, and simplicity of measurement, RDW and MPV are promising tools for risk assessment. Combined with other clinical parameters, these biomarkers could contribute to more accurate mortality prediction models in pediatric intensive care.

## Figures and Tables

**Figure 1 jcm-14-03839-f001:**
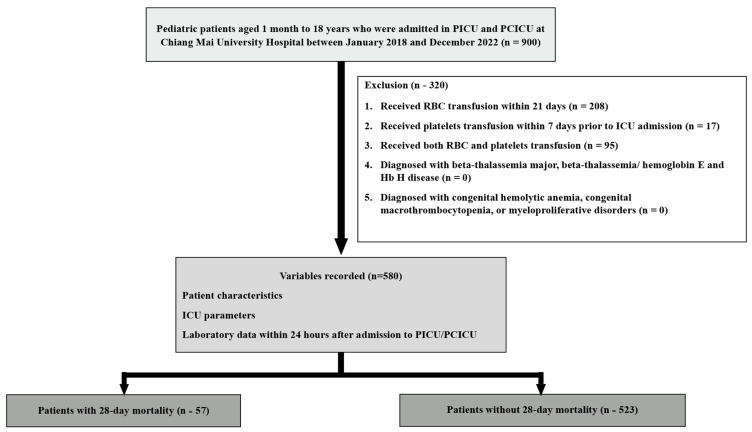
Flowchart of the study cohort.

**Figure 2 jcm-14-03839-f002:**
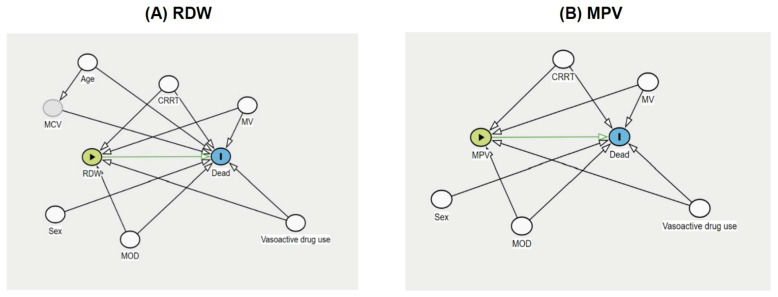
The causal diagram illustrates the predictive relationship of (**A**) RDW and (**B**) MPV with 28-day PICU mortality.

**Figure 3 jcm-14-03839-f003:**
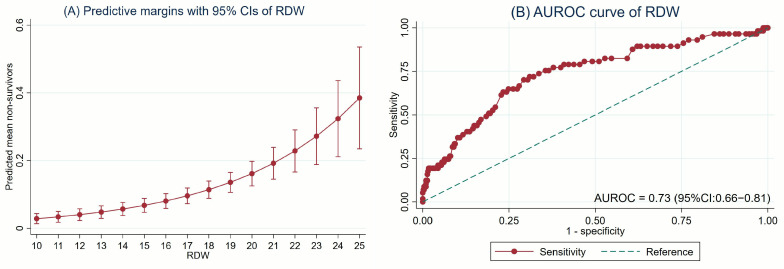
The elevated RDW levels with 28-day PICU mortality: (**A**) predictive margins with 95% CIs and (**B**) AuROC curve.

**Figure 4 jcm-14-03839-f004:**
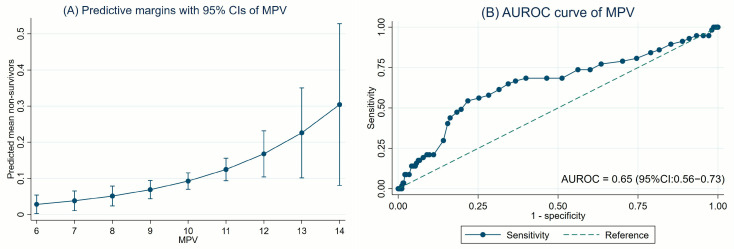
The elevated MPV levels with 28-day PICU mortality: (**A**) predictive margins with 95% CIs of MPV; (**B**) AuROC curve.

**Table 1 jcm-14-03839-t001:** Baseline characteristics of critically ill pediatric patients who survived and did not survive.

Characteristics	Total (*n* = 580)	Non-Survivors (*n*= 57)	Survivors (*n*= 523)	*p*-Value
Age (years)	5.9 (4.7, 10.4)	5.8 (4.7, 8.8)	6 (4.8, 10.4)	0.689
Male, n (%)	321 (55.3)	38 (66.7)	283 (54.1)	0.070
Body mass index (kg/m^2^)	15.2 (13.5, 17.4)	15.1 (12.9, 17.8)	15.2 (13.6, 17.4)	0.614
Comorbidities				
Seizure/epilepsy, n (%)	49 (8.4)	5 (8.7)	44 (8.4)	0.926
Chronic lung disease, n (%)	43 (7.4)	8 (14.0)	35 (6.7)	0.045
Cardiac arrythmia, n (%)	27 (4.6)	3 (5.3)	24 (4.6)	0.819
Asthma, n (%)	12 (2.1)	0 (0)	12 (2.3)	0.248
Neuromuscular disease, n (%)	1 (0.2)	0 (0)	1 (0.2)	0.741
Solid tumor, n (%)	22 (3.8)	0 (0)	22 (4.2)	0.114
Systemic lupus erythematosus, n (%)	10 (1.7)	2 (3.5)	8 (1.5)	0.276
Reason for intensive care unit admission				
Respiratory failure and acute respiratory distress syndrome, n (%)	418 (72.1)	51 (89.5)	367 (70.2)	0.002
Congestive heart failure, n (%)	130 (22.4)	20 (35.1)	110 (21.0)	0.016
Sepsis/Septic shock, n (%)	99 (17.1)	15 (26.3)	84 (16.1)	0.051
Post-operative care, n (%)	113 (19.5)	7 (12.3)	106 (20.3)	0.148
Neurologic condition/alteration of consciousness, n (%)	27 (4.6)	3 (5.3)	24 (4.6)	0.819
Others, n (%)	62 (10.7)	7 (12.3)	55 (10.5)	0.682
Illness severity				
Multi-organ failure, n (%)	259 (44.6)	46 (80.7)	213 (40.7)	<0.001
Pediatric Index of Mortality 2 score	2.9 (1.0, 6.6)	4.1 (1.5, 9.2)	2.8 (1.0, 6.3)	0.021
Management				
Mechanical ventilation, n (%)	432 (74.5)	54 (94.7)	378 (72.3)	<0.001
Vasoactive drugs, n (%)	217 (37.4)	45 (78.9)	172 (32.9)	<0.001
Continuous renal replacement therapy, n (%)	25 (4.3)	7 (12.3)	18 (3.4)	0.002

Continuous data are presented as median and interquartile range (IQR).

**Table 2 jcm-14-03839-t002:** Laboratory parameters of pediatric patients who survived and did not survive.

Laboratory Parameters	Total (*n* = 580)	Non-Survivors (*n* = 57)	Survivors (*n* = 523)	*p*-Value
Hemoglobin (g/dL)	11.2 ± 2.8	11.2 ± 3.4	11.6 ± 2.7	0.862
Hematocrit (%)	34.1 ± 8.7	34.7 ± 11.1	33.9 ± 8.4	0.559
White blood cell count (×10^3^ cell/mm^3^) *	12.0 (8.8, 16.6)	12.1 (7.7, 19.5)	12.0 (9.0, 16.5)	0.994
Absolute neutrophil count	7883.8 (4776.6, 11,733.8)	8020.2 (3689.6, 13,156.6)	7882.5 (4833.2, 11,707.5)	0.878
Absolute lymphocyte count	2590.9 (1551.4, 4187.5)	2333.4 (1210.7, 3695.1)	2592.8 (1580.6, 4218.4)	0.265
Platelet count (×10^3^ cell/mm^3^) *	285 (181, 400)	196 (86, 340)	292 (194, 408)	<0.001
Red blood cell distribution width (%)	15.7 ± 2.8	18.2 ± 3.6	15.5 ± 2.5	<0.001
Mean corpuscular volume (fL)	78.2 ± 9.1	81.1 ± 7.9	77.9 ± 9.2	0.014
Red blood cell count (×10^3^ cell/mm^3^)	4.4 ± 1.2	4.3 ± 1.4	4.4 ± 1.1	0.633
Nucleated red blood cell (%) *	0 (0, 0.1)	0.1 (0, 1.7)	0 (0, 0.1)	<0.001
Mean platelet volume (fL)	9.9 ±1.0	10.4 ± 1.2	9.8 ± 0.9	<0.001
Platelet distribution width (PDW)	10.8 ± 2.4	12.5 ± 3.0	10.7 ± 2.2	<0.001
Erythrocyte sedimentation rate (mm/h) *	24 (9, 41)	33 (16, 53)	24 (9, 39)	0.297
C-reactive protein (mg/L) *	15.5 (4.6, 62.5)	36.6 (7.9, 113)	15.1 (4.2, 62.4)	0.323
Lactate (mmol/L) *	2.2 (1.3, 3.7)	3.4 (2.1, 4.5)	2.1 (1.3, 3.4)	0.004
Procalcitonin (ng/mL) *	0.8 (0.2, 5.9)	1.3 (0.4, 18.0)	0.7 (0.2, 4.9)	0.087
Blood urea nitrogen (mg/dL) *	10 (6, 16)	13 (7, 27)	9 (6, 15)	0.002
Creatinine (mg/dL) *	0.3 (0.2, 0.5)	0.5 (0.3, 0.8)	0.3 (0.2, 0.4)	<0.001
Albumin (g/dL)	3.6 ± 0.7	3.4 ± 0.7	3.6 ± 0.7	0.084
Aspartate transaminase (U/L) *	41 (28, 79)	56.5 (36, 169)	40 (27, 73.5)	<0.001
Alanine transaminase (U/L) *	23.5 (14, 46)	31.5 (19, 86)	23 (14, 43)	0.010
Total bilirubin (mg/dL) *	0.4 (0.2, 0.9)	0.7 (0.3, 1.9)	0.4 (0.2, 0.8)	0.023
Direct bilirubin (mg/dL) *	0.2 (0.1, 0.5)	0.3 (0.1, 0.8)	0.2 (0.1, 0.4)	0.011

Continuous data are presented as mean ± SD. Otherwise (*) denotes a report as median and interquartile range (IQR).

**Table 3 jcm-14-03839-t003:** Univariable and multivariable risk regression analysis of RDW and MPV for predicting 28-day ICU mortality in the critically ill pediatric patients.

Parameters	Univariable Analysis		Multivariable Analysis	
	Unadjusted RR (95% CI)	*p*-Value	Adjusted RR (95% CI)	*p*-Value
RDW (%)	1.25 (1.19–1.33)	<0.001	1.19 (1.13–1.25)	<0.001
RDW ≥ 15% *	3.82 (2.02–7.23)	<0.001	2.73 (1.45–5.13)	0.002
MPV (fL)	1.54 (1.25–1.89)	<0.001	1.34 (1.10–1.64)	0.003
MPV ≥ 10 fL *	3.04 (1.79–5.15)	<0.001	2.38 (1.43–3.93)	0.001

* Adjusted for age, sex, use of MV, use of vasoactive drug, use of CRRT, and MOD.

**Table 4 jcm-14-03839-t004:** Diagnostic performance of RDW and MPV for determining 28-day PICU mortality.

Parameters	Non-Survivors (*n* = 57)	Survivors (*n* = 523)	Sensitivity	Specificity	PPV	NPV	AuROC
RDW ≥ 15%	46 (80.7)	257 (49.1)	80.7%	50.9%	15.2%	96.0%	0.66 (0.60–0.71)
MPV ≥ 10 fL	38 (66.7)	192 (36.7)	66.7%	63.3%	16.5%	94.6%	0.65 (0.58–0.71)

AuROC: area under the receiver operating characteristic; MPV: mean platelet volume; NPV: negative predictive value; PPV: positive predictive value; RDW: red blood cell distribution width.

## Data Availability

The raw data supporting the conclusions of this article will be made available by the authors on request.

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
