# Peer review of "Admission Red Blood Cell Distribution Width and Mean Platelet Volume as Predictors of Mortality in the Pediatric Intensive Care Unit: A Five-Year Single-Center Retrospective Study"

_jcm, 2025, doi:10.3390/jcm14113839_

Round 1

Reviewer 1 Report

Comments and Suggestions for Authors

The manuscript presents a retrospective analysis of 580 patients at a pediatric intensive care unit (i.e. no experimental design or targeted interventions). However, these data are well prepared. The study confirmed data from various previous studies [1,2,5,13,14,24]. No additional mechanistic insight was generated.

Author Response

We sincerely appreciate your thoughtful comments and the time you have taken to review our manuscript.

Reviewer 2 Report

Comments and Suggestions for Authors

I read with much interest the paper entitled “Can Red Blood Cell Distribution Width and Mean Platelet Volume Predict Mortality in the Pediatric Intensive Care Unit?” by Sunkonkit and colleagues. The study is very interesting; indeed, studying risk factors for mortality in critically ill patients helps to adopt increasingly targeted and effective strategies to improve the outcome of these patients. The study compares RDW and MPV values between survivors and non-survivors and highlights the following:

  • The non-survivor group demonstrated a significantly prolonged duration of MV (p < 0.001) and extended use of vasoactive medications (p < 0.001) compared to the survivor group. Additionally, the non-survivors exhibited significantly higher levels of RDW (p < 0.001), mean corpuscular volume (p = 0.014), MPV (p < 0.001), and platelet distribution width (PDW) (p < 0.001), along with a notably lower platelet count (p < 0.001).

Congratulations to the authors for their excellent work. However, I have some questions to ask.

The statistically significant difference of the two values is shown in groups of patients in whom vasoactive drug treatment and prolonged mechanical ventilation are present; since these risk factors are independently associated with mortality how do these new parameters account for increased mortality risk and how are mechanical ventilation related to the other factors? In fact, prolonged mechanical ventilation and vasoactive support are treatments frequently used in pathological conditions such as ARDS and septic shock, burdened by high mortality, and the activation of the inflammatory response in these pathologies could explain the increase in RDW and MPV values, so these would not represent a risk factor but simply a laboratory manifestation of these pathologies. Therefore, it would be fair to discuss these points raised perhaps by expanding the limitations paragraph. Otherwise I think the paper is really well done and can be published after arguing on the points raised.

Author Response

#Comment 1: The statistically significant difference of the two values is shown in groups of patients in whom vasoactive drug treatment and prolonged mechanical ventilation are present; since these risk factors are independently associated with mortality how do these new parameters account for increased mortality risk and how are mechanical ventilation related to the other factors? In fact, prolonged mechanical ventilation and vasoactive support are treatments frequently used in pathological conditions such as ARDS and septic shock, burdened by high mortality, and the activation of the inflammatory response in these pathologies could explain the increase in RDW and MPV values, so these would not represent a risk factor but simply a laboratory manifestation of these pathologies. Therefore, it would be fair to discuss these points raised perhaps by expanding the limitations paragraph. Otherwise, I think the paper is really well done and can be published after arguing on the points raised.

Response: We sincerely appreciate the reviewer’s insightful and constructive comments. We agree with the reviewer that vasoactive drug use and prolonged mechanical ventilation are well-established independent predictors of mortality in critically ill pediatric patients. In our cohort, elevated RDW and MPV levels were indeed observed more frequently in these subgroups, reflecting the severity of the underlying disease processes such as ARDS and septic shock, both of which are characterized by systemic inflammation and higher mortality risk. We recognize that RDW and MPV may not act as independent causative risk factors for mortality but rather as surrogate markers of disease severity and systemic inflammatory response.

In response to this important concern, we have revised the Discussion section to emphasize that RDW and MPV are also indicators of critical illness severity. Furthermore, we have expanded the Limitations section to address potential confounding factors, including disease severity and related interventions such as mechanical ventilation and vasoactive therapy.

The updated version is as follows:

Discussion (page 12, line 368-387)

“Moreover, elevated RDW and MPV values were significantly associated with mortality in the PICU in our study. However, it is essential to interpret these findings within the clinical context. Patients who required prolonged MV or vasoactive support demonstrated higher RDW and MPV levels, which likely reflect the underlying sever-ity of their critical illness rather than indicating that these hematologic parameters are independent prognostic factors. Conditions such as ARDS and septic shock are com-monly accompanied by systemic inflammatory responses, which can induce bone marrow stress, ineffective erythropoiesis, and thrombopoiesis abnormalities, thereby elevating RDW and MPV values[44,45]. Thus, these laboratory abnormalities may also serve as markers of severe inflammatory burden. Our findings underscore the im-portance of considering RDW and MPV within the broader clinical picture, integrating these parameters alongside established indicators of illness severity to guide risk strat-ification and clinical decision-making in PICU.

Notably, RDW demonstrated better discriminatory ability for predicting 28-day PICU mortality in our cohort compared to the PIM-2 score. In contrast, MPV did not exhibit a statistically significant difference in discriminative performance when com-pared with the PIM-2 score. Nevertheless, MPV presents a more practical advantage, as it is easier to obtain and interpret, relying on a single parameter rather than the multiple variables required for calculating the PIM-2 score. However, further valida-tion studies are warranted to confirm these findings.”

Discussion (limitation part) (page 13, line 400-404)

“Third, RDW and MPV elevations may reflect the severity of underlying disease states, such as ARDS and septic shock, which are themselves associated with high mortality rates and require interventions like MV and vasoactive support. These factors may confound the observed associations, as both RDW and MPV could act as surrogate markers rather than independent predictors of mortality.”

Reviewer 3 Report

Comments and Suggestions for Authors

This is an interesting study—congratulations. However, several modifications are necessary to enhance the scientific rigor and clarity of the manuscript.

1.The title should include details regarding the type of study (e.g., single-center study) and the timing of the analysis (e.g., data collected at admission).

Abstract

2.Line 25: appears 15 years as maximal age limit (you used 18)

Introduction

3.The manuscript should include a theoretical rationale for the correlation between the selected parameters and mortality, to help readers understand the basis for their inclusion in the study.

Material and methods

4.Line 105: Why 28-day mortality, you should address this somewhere in the manuscript. (e.g. 28 days correspond to exactly four weeks, which aligns well with clinical follow-up schedules, simplifies data collection intervals, and facilitates consistent time framing in statistical analyses).

Results

Figure 1: the text for me seems to be with too small characters

It would be beneficial to correlate the selected parameters with established mortality prediction scores (e.g., APACHE II), assuming the relevant data are available, as this could strengthen the validity of your findings.

Author Response

Reviewer #3: 

This is an interesting study—congratulations. However, several modifications are necessary to enhance the scientific rigor and clarity of the manuscript.

#Title:

1.The title should include details regarding the type of study (e.g., single-center study) and the timing of the analysis (e.g., data collected at admission).

Response: Thank you for your suggestion. We have already edited the title of the manuscript (page 1, line 2-4) to “Admission Red Blood Cell Distribution Width and Mean Platelet Volume as Predictors of Mortality in the Pediatric Intensive Care Unit: A Five-Year Single-Center Retrospective Study”.

#Abstract

2.Line 25: appears 15 years as maximal age limit (you used 18)

Response: Thank you for your interesting point. We have already edited the Abstract (method part) as below.

Abstract (page 1, line 25-28)

We have edited as per the reviewer's suggestion.

Methods: This retrospective cohort study analyzed data from children aged 1 month to 18 years who were admitted to the PICUs at Chiang Mai University Hospital for ≥24 hours between January 2018 and December 2022.”

#Introduction

3.The manuscript should include a theoretical rationale for the correlation between the selected parameters and mortality, to help readers understand the basis for their inclusion in the study.

Response: Thank you for your valuable suggestion. We have revised the Introduction to incorporate a theoretical rationale explaining the correlation between the selected hematological parameters (RDW and MPV) and mortality, to enhance the readers' understanding of their inclusion in the study. Specifically, we have expanded the Introduction to provide a more comprehensive explanation of the underlying pathophysiological mechanisms, as reflected in the revised text (page 2, lines 55–73).

“RDW has traditionally been utilized as an ancillary marker to aid in differentiating the causes of anemia[9]. However, emerging evidence has identified RDW as a significant prognostic indicator across various systemic diseases, with increasing recognition of its association with mortality in critically ill patients[9-14]. MPV, derived by dividing the platelet-crit (PCT) by the platelet count [15,16], reflects the average size of circulating platelets and is readily measured through automated hematology analyzers. Elevated MPV is frequently observed in thrombotic and inflammatory states, reflecting heightened platelet activation and function [17,18].

Both RDW and MPV are readily accessible laboratory parameters that encapsulate underlying pathophysiological changes common in critical illness. RDW, an indicator of anisocytosis, has been linked to systemic inflammation, oxidative stress, and impaired erythropoiesis — conditions prevalent among critically ill children and associated with adverse clinical outcomes[19,20]. Similarly, MPV serves as a surrogate marker of platelet activation, which plays a central role in the inflammatory cascade and the development of microthrombotic complications seen in severe infections and systemic inflammatory states[18]. Prior studies have demonstrated that elevated MPV levels correlate with poor prognosis in both adult and pediatric critically ill populations[8]. Collectively, these mechanisms provide a compelling biological rationale for evaluating RDW and MPV as potential prognostic indicators of mortality risk in the pediatric intensive care unit (PICU).”

#Material and methods

4.Line 105: Why 28-day mortality, you should address this somewhere in the manuscript. (e.g. 28 days correspond to exactly four weeks, which aligns well with clinical follow-up schedules, simplifies data collection intervals, and facilitates consistent time framing in statistical analyses).

Response: Thank you for your insightful comment. We have clarified the rationale for selecting 28-day mortality as the study endpoint in the revised manuscript. Specifically, the 28-day time frame corresponds to a full four-week period, which aligns well with typical clinical follow-up schedules in the PICU setting, facilitates consistent data collection intervals, and provides a standardized time frame commonly used in critical care research for outcome measurement. This approach ensures comparability with previous studies and enhances the interpretability of our findings. The explanation has been added to the Materials and Methods section (page 3, lines 117-121).

“28-day PICU mortality was defined as death occurring within 28 days of admission to the PICU. This 28-day period represents a complete 4-week interval, which aligns with standard clinical follow-up practices in the PICU setting, ensures uniformity in data collection intervals, and corresponds to a widely accepted timeframe for outcome assessment in critical care research.”

#Results:

5.Figure 1: the text for me seems to be with too small characters

Response: We appreciate your helpful suggestion. The characters in Figure 1 have been enlarged to enhance readability, as updated in the revised manuscript (page 4).

6. It would be beneficial to correlate the selected parameters with established mortality prediction scores (e.g., APACHE II), assuming the relevant data are available, as this could strengthen the validity of your findings.

Response: Thank you very much for your valuable suggestion. We fully concur that correlating RDW and MPV with established mortality prediction scores, such as APACHE II, would strengthen the validity of our findings. However, we regret to inform you that APACHE II scores were not routinely recorded in our PICU during the study period. Instead, we employed the Pediatric Index of Mortality 2 (PIM-2), which is the standard mortality prediction tool utilized in our unit and is widely validated for use in critically ill pediatric populations.

Our analysis revealed that the PIM-2 score was significantly associated with 28-day PICU mortality, with a RR of 1.02 (95% CI: 1.02–1.03; p < 0.001). The AuROC was 0.59 (95% CI: 0.51–0.68). We have incorporated these results into the revised Results section (page 9, line 240-243) and presented them in the newly added Figure 5A and 5B, following the additional analysis of the association between PIM-2 and PICU mortality. Furthermore, we have expanded the Discussion section (page 12, line 381-387) to address this point and to further strengthen the interpretation and validity of our findings.

Results (page 9, line 240-243)

“The PIM-2 score was associated with 28-day PICU mortality, yielding an Au-ROC of 0.59 (95% CI: 0.51–0.68). Interestingly, RDW demonstrated superior predictive performance for 28-day PICU mortality compared to the PIM-2 score (p = 0.015), whereas MPV showed no statistically significant difference when compared with PIM-2 score (p = 0.38).”

Discussion (page 9, line 245-256)

“Our study underscores the significant association between elevated RDW and MPV levels, measured within 24 hours of PICU admission, and 28-day PICU mortality. Although the AuROC analyses for RDW and MPV, as standalone predictors, indicated only moderate clinical utility when applying a single cut-off point, their discriminative performance improved markedly when patients were stratified into low- and high-risk groups. Specifically, RDW and MPV thresholds of 15% and 10 fL, respectively, yielded high NPV for 28-day PICU mortality, at 96.0% for RDW and 94.6% for MPV.

Importantly, as both RDW and MPV are routinely obtained from a standard CBC, they represent cost-effective and widely accessible tools for early risk stratification. These parameters might be integrated with other clinical variables to develop more comprehensive predictive models for mortality in critically ill pediatric patients, thereby supporting more informed clinical decision-making and potentially improving patient outcomes.”

and Discussion (Page 12, line 381-387)

“Notably, RDW demonstrated better discriminatory ability for predicting 28-day PICU mortality in our cohort compared to the PIM-2 score. In contrast, MPV did not exhibit a statistically significant difference in discriminative performance when com-pared with the PIM-2 score. Nevertheless, MPV presents a more practical advantage, as it is easier to obtain and interpret, relying on a single parameter rather than the multiple variables required for calculating the PIM-2 score. However, further validation studies are warranted to confirm these findings.”